# Conservative and Pharmacological Strategies for Preventing Osteoporotic Stress Fractures in Older Recreational Competitors

**DOI:** 10.3390/healthcare13182328

**Published:** 2025-09-17

**Authors:** Lana Ružić, Marija Rakovac, Ines Bilić-Ćurčić, Domagoj Jakovac, Maja Cigrovski Berković

**Affiliations:** 1Faculty of Kinesiology, University of Zagreb, 10110 Zagreb, Croatia; lana.ruzic.svegl@kif.unizg.hr (L.R.); marija.rakovac@kif.unizg.hr (M.R.); maja.cigrovski.berkovic@kif.unizg.hr (M.C.B.); 2Faculty of Medicine, University J.J. Strossmayer, 31000 Osijek, Croatia; ibcurcic@mefos.hr; 3School of Dental Medicine, University of Zagreb, 10000 Zagreb, Croatia

**Keywords:** antiresorptive agents, bone density, older athlete, prevention, sports injury

## Abstract

Exercise and bone health are crucial for overall cardiovascular and metabolic well-being. However, certain types of activities can increase the risk of osteoporotic fractures. Preventing injuries while remaining active can be challenging, particularly for older competitive recreational athletes whose training volumes can be comparable to those of professional athletes. While high levels of physical activity in older adults typically lead to thicker cortical bone and improved physical fitness, age-related bone loss, hormonal changes, nutrition, and a history of fractures can significantly raise the risk of osteoporosis. This narrative review provides a descriptive summary of the current understanding of the paradox between exercise and bone fragility, particularly in recreational athletes. Many older athletes may be unaware of their declining bone mineral density, believing that their activity levels are sufficient. Ironically, high-impact activities like running and jumping, which are generally recommended for the prevention of osteoporosis, can also increase the risk of stress fractures. Early detection of osteoporosis is crucial, and this can be achieved through DEXA scans and regular bone density tests, allowing for timely intervention before fractures occur. Alongside medications, strength training, as well as balance and stability exercises, can be very beneficial. It is also important to maintain healthy lifestyle choices, ensure adequate energy levels, and consume sufficient amounts of calcium and vitamin D. Furthermore, monitoring training volume and allowing for proper recovery can help reduce the risk of osteoporotic stress fractures in athletes.

## 1. Introduction

Osteoporosis weakens bones, making them fragile and more prone to fractures. It is more common in individuals over the age of 50 and affects both men and women. However, postmenopausal women are at a higher risk, which is why there are specific clinical guidelines for osteoporosis screening in this group. Unfortunately, similar guidelines are still lacking for the male population [1]. Exercise and bone health are essential for overall cardiovascular and metabolic well-being. However, some forms of activity can increase the risk of osteoporotic fractures, making it challenging to prevent injuries while staying active. The number of recreational athletes is on the rise. Although they are not professionals, the frequency, intensity, and volume of their training often match that of competitive athletes. In this review, the term “recreational competitor” refers to older adults who consistently participate in structured sports training and competitions, even though they are not classified as elite or professional athletes. Activities such as running, skiing, tennis, cycling, and soccer are amongst the most popular ones, and although they promote heart health, increase muscle strength, and positively impact mental well-being, they also put stress on bones and joints, and at the same time help maintain bone mineral density (BMD). Conversely, high-impact activities or high volume and intensity training without adequate recovery can even increase the risk of fractures, especially in those who already have osteoporosis [2]. This paper addresses a growing and less-studied population of aging recreational competitors. From both scientific and practical perspectives, this group is often overlooked in discussions surrounding osteoporosis risk management. For recreational athletes, finding a healthy balance between the positive effects of exercise and its associated risks is crucial.

As people live longer, a significant proportion of older adults are staying active by participating in recreational sports and exercise. For instance, hundreds of women over the age of 70 and thousands over the age of 60 participated in last year’s New York Marathon, with even more running marathons worldwide. Although the exact number of noncompetitive recreational runners is unknown, it is steadily increasing. The average age of marathon runners who complete all six “big” marathons is around 50, while some participants are in their late seventies.

Research shows that high levels of physical activity in older adults are typically associated with thicker cortical bone, improved physical function, and a reduced risk of fractures, even if other risk factors are present [3]. It is important not to assume that aging athletes are completely protected from osteoporosis simply because they remain active. Several factors, such as age-related bone loss, changes in hormone levels, nutrition, and a history of previous fractures, can significantly increase the risk of osteoporosis. As people age, bone remodeling slows down, and the decreasing concentrations of estrogen and testosterone can impair bone mineralization. Additionally, recreational athletes who engage in high-impact activities may experience repetitive microinjuries that can lead to irreparable damage, resulting in stress fractures, especially if accompanied by poor nutrition and low vitamin D levels. Unfortunately, in the older population, bones become less responsive to mechanical loading, and the remodeling process slows down due to reduced activity of osteoblasts. Hormonal changes also contribute to this process. These changes can impair the ability of the bone to adequately repair the microdamage that naturally occurs with repetitive stress. Exercise is well known to improve bone density and prevent vertebral fractures [4]. Nevertheless, fractures related to osteoporosis often occur in the extremities due to stress and high ground reaction forces experienced during intense physical activities such as running. These fractures are commonly found in weight-bearing bones of the pelvis and legs. In contrast, upper-body stress fractures usually result from muscle forces on the upper torso and are less common [5]. In a Matheson et al. paper, the authors reported that in 320 athletes, lower leg fractures were the most common, occurring in 49.1% of cases, followed by fractures of tarsal and metatarsal bones. The femur and fibula were the least affected bones. Repetitive impact from running and jumping significantly contributes to a high risk of fractures [6]. This would likely be even more apparent in older athletes.

Many aging athletes do not perceive the declining bone mineral density as they consider themselves to be sufficiently active. This perception may lead them to train through minor injuries and discomfort, which can escalate to more serious injuries. In case they fail to make changes in their training routine, nutrition, or consider pharmacological options, they risk an early end to their sports career and even loss of mobility [7]. Therefore, in ageing athletes, regular bone density screenings are essential. In addition, it is important to modify frequency, volume, and training intensity. Regardless of the amount of aerobic workout, such as that performed by runners, weight-bearing exercises, which improve bone health, are crucial [1,8,9].

Recreational athletes with developed osteoporosis who continue to engage in high-intensity activities are at a higher risk of stress injuries. Unfortunately, it can be difficult for these active individuals to accept that their rigorous exercise routines did not have a healthy impact on their bones. As a result, they may be reluctant to lower the volume and intensity of their usual training while they consider the exercise of utmost importance for maintaining functional and motor abilities as well as overall health. However, high-impact activities such as running or heavy gym workouts might easily speed up bone microdamage and increase the fracture risk [7].

If an osteoporotic fracture occurs, it can result in chronic pain, loss of flexibility, even mobility issues, and ultimately in a reduced quality of life. Experiencing just one osteoporotic fracture carries a higher risk of future fractures. Hip fractures, in particular, are serious injuries that often result in a complete loss of independence, which can be devastating for an older athlete who was previously active.

The primary goal of this review is to inform and alert all experts and organizers involved in recreation and sporting activities for older participants. It is crucial to recognize that the training practices typically used for younger athletes should not be applied in the same way to this population due to differences in human biology. Additionally, the review emphasizes the importance of effectively managing potential risks.

## 2. Methods

The sources for this review were identified from several databases, including Scopus, Web of Science (WoS), Sport Discus, and PubMed. This article presents a narrative review that synthesizes existing evidence on osteoporotic stress fractures and prevention strategies specifically for older recreational athletes. We prioritized articles published after 2010; however, we also included older studies, particularly those discussing mechanisms of bone remodelling. The goal of this review is to summarize existing knowledge applicable to the older athletic population and to identify specific challenges and potential therapeutic options for this group. The key terms used in our search were: “osteoporosis,” “stress fracture,” “older athlete,” “recreational athlete”, “bone remodelling,” “antiresorptive agents,” and “fracture prevention.” Most of the selected articles focused on bone health in older athletes or the risk of fractures. This review is descriptive and summative in nature, and no statistical synthesis or original data analysis was performed.

## 3. How Does Aging Affect Bone Health

Human bones constantly undergo remodeling. In younger individuals, bone formation tends to exceed bone loss, resulting in strong and dense bones. Most people reach their peak bone mass around the age of 30 [10,11]. After this age, the rate of bone loss can vary, depending on factors such as age and lifestyle. Strength training and activities that generate impact from ground reaction forces, like walking, jogging, and running, may help slow down bone loss. However, not all activities are equally beneficial. Those involving high impacts, like jumping, place additional stress on bones and promote remodeling. In contrast, non-weight-bearing activities, such as cycling and swimming, which are popular among older adults, do not provide enough stress to increase bone density [12]. Older athletes often face additional challenges, such as degenerative joint disease, which are often alleviated by cycling and swimming [13,14]. Although the mentioned activities improve cardiovascular health and prevent metabolic diseases, their impact on bone health is minimal, especially in critical fracture-prone sites like the spine and femoral neck. Additionally, women are at higher risk as they tend to lose bone density more rapidly after menopause due to estrogen deficiency [15]. It does not mean that men are protected; they also lose bone mass, but at a somewhat slower pace. In older masters or recreational athletes, the aging skeleton has diminished sensitivity to mechanical loads and also slower turnover, which significantly reduces physiological responses and remodeling, essentially minimizing anabolic adaptations.

### 3.1. Exercise and Fracture Risk Paradox

Although exercise is essential for bone health, some types of activities can increase the risk of fractures in aging athletes, particularly those with developed low bone density or osteoporosis. This situation creates a paradox that can be explained as follows: the normal bone remodelling process is enhanced by the “push and pull” forces generated by muscles acting on the skeleton. Commonly recommended exercises for preventing osteoporosis and stress fractures typically involve high-impact activities such as jumping or running. However, in individuals with weaker bones, these activities can lead to small architectural imperfections in the bone, especially among older athletes. This is largely because the anabolic response to these forces is lower in older individuals compared to younger populations. Paradoxically, in that case, the prevention treatments could contribute to the occurrence of small, hardly visible stress fractures, but if the athlete is unaware of those fractures or underestimates the pain and does not take enough recovery time, these can result in larger fractures [16].

Falls are another major cause of fractures in older adults, primarily due to declines in muscle strength, balance, and coordination. Due to the ageing of the neurological system, the coordination between the brain and muscles weakens. Even physically active older athletes, if they neglect specific stability exercises, can have balance problems. Fractures, particularly of the hip and wrist, usually occur as a result of falls [17].

### 3.2. Risk Factors for Osteoporotic Fractures in Older Athletes

#### 3.2.1. Hormonal Imbalances

As people become older, their bones naturally lose strength, making them more prone to breaking. This is mainly due to a gradual loss of BMD, which increases the risk of fractures, especially in the weight-bearing areas like the femoral neck, spine, and wrists (in case of falls) [18]. The body’s ability to build new bone slows down with age, and as the breakdown continues, osteopenia and osteoporosis develop. The latter is mainly seen in postmenopausal women, due to estrogen loss, but it should not be underestimated in men [1].

Estrogen plays a key role in keeping bones strong, and when its levels decrease, bone loss occurs more rapidly. Women in menopause may protect against bone loss to some extent by taking hormone replacement therapy (HRT), but this is not an option for everyone because of the potential risks, such as an increased chance of blood clots and the possibility of certain cancers [19]. Testosterone in men is also important for the maintenance of bone strength. With ageing, lower testosterone levels do contribute to osteoporosis, but at a slower pace than seen in women with estrogen loss [20]. An additional issue to consider in an aging athletic population is the use of glucocorticoid agents, often prescribed due to their effect in managing acute inflammatory and musculoskeletal conditions. Depending on the cumulative dosage, age, and low body weight, they can lead to significant bone demineralization and increase the risk of bone fractures in older athletes [21]. Moreover, hyperthyroidism (both overt and clinically manifest), as a consequence of toxic multinodular goiter and toxic adenomas, which become more prevalent with age, or due to levothyroxine treatment-induced TSH suppression, might reduce BMD, especially in postmenopausal women, and increase the risk of fractures [22].

#### 3.2.2. Intake of Nutrients

To maintain BMD, the body needs proper nutrition, primarily enough calcium and vitamin D. While calcium is necessary for bone structure, vitamin D helps the body absorb calcium more efficiently. Deficiencies in either nutrient are common among older adults [23] and are strongly associated with increased risk of fracture [24,25].

Low calcium intake leads to high fracture risk [22], and if its absorption is not enabled by enough vitamin D, as seen in the case of a lack of sunlight exposure, the risk increases [25]. Although we would expect older athletes to lead and maintain a health-oriented lifestyle, they still underestimate their nutritional needs during periods of high-volume and high-intensity training. Therefore, it is important to regularly evaluate calcium and vitamin D intake, preferably before the conditioning phase of the season and once during the competitive period (typically twice a year for most sports). Based on laboratory findings and nutritional questionnaires, supplementation should be considered as needed.

#### 3.2.3. Previous Fractures

Any athlete who has already suffered an osteoporotic fracture may be more prone to a future fracture. In women, an early warning sign of osteoporosis may be a wrist fracture, while such a fracture increases the risk of spine fractures later in life [26,27]. However, in athletes and runners, not every wrist fracture poses the risk for future fractures, as some are merely a result of an incidental fall. If more serious stress fractures occur, e.g., small vertebral fractures, it will cause chronic pain resulting in a hunched posture, which will usually make individuals withdraw from amateur sport. In athletes, hip fractures can be particularly serious, as they often lead to loss of mobility, long hospital stays, and even an increased risk of death [28,29,30]. Interestingly, there are currently no clear statistics on the incidence of hip fractures among marathon runners, indicating a need for future research in this area. The occurrence of any fracture in athletes over 50, especially under low volume, necessitates urgent assessment of bone health using radiographic methods, DEXA scans, and laboratory tests.

#### 3.2.4. Proper Recovery

Recovery time between strenuous activities, especially running, may also be a vital factor in preventing osteoporotic fractures in older athletes. Certain exercises and sports can increase the risk of fractures in older adults, particularly those with osteoporosis. High-impact activities place excessive stress on bones, and activities such as running, jumping, and other repetitive high-impact exercises can cause stress fractures, especially if athletes do not allow enough recovery time between workouts. For example, there is a possible link between hepcidin deficiency and higher bone loss by disruption of the Wnt/β-catenin signaling pathway through the activation of FOXO3a transcription factor, which shifts stem cells away from their role in bone formation [31,32]. Hepcidin is a hormone that regulates iron levels and also contributes to bone metabolism. Typically, hepcidin levels increase after high-intensity exercise, which aids in recovery. Research indicates that in postmenopausal female athletes, hepcidin levels may remain suppressed for extended periods. This delayed recovery of hepcidin can hinder bone formation by disrupting the Wnt/β-catenin pathway, which is crucial for activating osteoblasts and facilitating bone repair. Moreover, the high-impact activities that are intended to enhance bone remodelling during recovery can lead to increased microdamage in the bones. As a result, older athletes may require longer recovery periods to support effective bone remodelling. In our expert opinion, the commonly recommended 48 h recovery time should be extended to a minimum of 72 h, avoiding high-impact forces on the bones during this period.

Incorporating low magnitude and high frequency (LMHF) mechanical loading through activities such as postural control exercises can be beneficial. These exercises can be safely performed even during recovery from high-impact activities, as they have been shown to have a positive anabolic effect on bones [33].

#### 3.2.5. Low Energy Availability

The connection between osteoporosis and low energy availability (LEA) and relative energy deficiency in sport (RED-S) is well established. There is a significant variation in the prevalence of LEA/RED-S in different sports (reported prevalence range of 23–79.5% for females and 15–70% for males). However, the highest prevalence is reported for endurance athletes [34,35]. In the studies addressing the energy availability in middle- and long-distance runners, LEA was identified in 43.4% of the athletes [36]. In endurance sports like marathons and long-distance running, the high volume of activity may contribute to low energy availability, leading to RED-S. LEA can, thus, be unintentional if the athletes are unaware of the energy demands imposed by their training program, or intentional, if the athletes deliberately restrict their energy intake [34]. Adequate dietary intake for older athletes depends on energy needs from their sport and is influenced by age-related physiological and metabolic changes, along with potential risk factors or chronic illnesses [37]. A recent systematic review indicated that master athletes may not meet their dietary intake requirements necessary for both health and athletic performance [37]. At the 2022 and 2023 World Masters Athletics Championships, it was noted that competitors did not meet the recommended intake of carbohydrates, proteins, vitamins D, E, and K, calcium, potassium, vitamin A, and other nutrients [38].

With regard to bone health, short-term LEA has been linked with increased markers of bone resorption, i.e., decreased markers of bone formation in physically active females, while in men this relationship is less understood [39]. The short-term LEA effect on bone metabolism has, among others, been linked with decreased insulin and leptin levels [39]. As evidenced by cross-sectional studies, long-term LEA in physically active persons was related to lower bone mass, bone resorption, decreased bone strength, and increased stress fracture risk [35]. To avoid these negative consequences, the athletes—especially older ones—should have nutritional support to ensure energy availability of at least 30-45 kcal kg lean body mass^−1^ day^−1^ and the adequate intake of micronutrients essential for bone health, previously identified as deficient in this population, including calcium, vitamin D, and K [38,39].

#### 3.2.6. Managing Other Conditions and Lifestyle

Older athletes may also suffer from other medical conditions that contribute to the development of osteoporosis. Osteoporosis has been known to be more common in chronic conditions like diabetes, hypo- and hyperthyroidism, rheumatoid arthritis, and kidney diseases. As those conditions influence bone loss, they heighten the fracture risk, in the overall population as well as in older athletes [40]. Medications, especially corticosteroids, which were previously widely used to lower inflammation, weaken the bones [41]. Several lifestyle factors also play a role in bone density. Smoking reduces BMD and increases the risk of fractures, probably by affecting calcium absorption [42]. Like smoking, excessive alcohol consumption interferes with osteocytes and leads to weaker bones. No matter their healthier lifestyle, alcohol consumption should be suspected as a cause for osteoporosis and fractures in male athletes [43]. Maintaining a healthy weight is important, as both underweight and overweight individuals can have a higher risk of fractures. Being underweight is linked to lower bone mass, while being overweight can increase stress on joints and impair stability, making falls more risky [44].

## 4. Clinical Screening Tools

In a clinical setting, accurately assessing the severity of bone mineral density loss is essential for selecting appropriate therapies. While dual-energy X-ray absorptiometry (DXA) is considered the gold standard for measuring bone density, it has limitations, particularly in not accounting for various risk factors contributing to the risk of major osteoporotic fractures. To address this, the FRAX score has been introduced as an additional tool. Currently, a new FRAXplus score is being tested, including information on fall history, the presence of diabetes, and corticosteroid dose exposure, if applicable. This enhanced scoring system aims to improve the estimation of residual fracture risk, which may be especially relevant for aging athletes [45]. Furthermore, assessing the risk of osteoporosis should not be limited to females. The mortality rate associated with fragility fractures is even higher in men. Current guidelines recommend osteoporosis assessment for men starting at age 70; however, this approach may soon be reconsidered, especially within the athletic population [46,47].

A critical issue to consider is the biological longevity of athletes achieved by participation in sports. In terms of bone health, exercise is one of the most important stimuli for osteoanabolic response, especially for the BMD of the hip [48], where it exerts a protective role against fractures [49]. On the other hand, vertebral BMD might not be positively affected by exercise, and exercise, including twisting and bending, might provoke vertebral fractures [50]. Female athletes who started their sports career at a young age might be at special risk due to problems with body weight, irregular menstrual cycles due to hypoenergosis, and lower maximal bone mass, as well as the existence of previous bone trauma/microfractures due to the female athlete triad [51]. Effective management of primary fracture prevention is crucial and should likely begin earlier than in the general population, where attention to bone health typically starts at menopause.

For assessment of BMD in premenopausal women and men under 50, Z-scores compare bone density with age- and sex-matched controls, and a score < 2 with additional risk factors for fractures is recommended to diagnose osteoporosis. Athletes usually have a 10% to 15% higher BMD than nonathletes, so the American College of Sports Medicine defines ‘‘low BMD’’ as a history of nutritional deficiencies, hypoestrogenism, stress fractures, and/or other secondary clinical risk factors for fracture together with a Z-score between −1.0 and −2.0, and the osteoporosis as secondary clinical risk factors for fracture with BMD Z-scores less than −2.0 [51,52].

A DEXA scan is recommended for male recreational athletes starting at age 60. For females, it is suggested five years after menopause. Every 1–2 years is a recommended period to repeat the scan to monitor bone density changes. For the athletes over the age of 50, a DEXA scan should be considered if there is a history of fractures, particularly low-impact ones.

## 5. Application of Anti-Resorptive/Anabolic Agents in Older Athletes

Antiresorptive agents, bisphosphonates, are the first agents of choice for postmenopausal and senile osteoporosis for men and women. Data from randomized controlled trials (RCTs) suggest a positive impact on bone mineral density (BMD), particularly with ibandronate at vertebral sites, while risedronate, alendronate, and zoledronic acid are, in addition to vertebral fracture prevention, effective in reducing the risk of non-vertebral and hip fractures [53]. These agents might be considered in the case of a premenopausal athlete with osteoporosis and low response to hormone therapy, as well as to treat stress fractures [54]. Similarly, anabolic agents like teriparatide or abaloparatide could be used for up to two years in case of a very low BMD and fractures not responding to hormone therapy or delayed fracture healing, but the mentioned therapies are off-label [55,56]. Besides the lack of data from randomized controlled trials (RCTs), agents like bisphosphonates have a long half-life and teratogenic effects; therefore, they could pose a problem for an athlete of childbearing age [52].

For older professional athletes who are no longer competing, the treatment of osteoporosis requires a specific approach that balances their unique musculoskeletal history with age-related decline in bone metabolism. Although they may have achieved higher peak bone mass from years of training, the cessation of high-impact activity, along with a history of energy deficiency or steroid use [35] and age-related hormonal changes, such as reduced estrogen in postmenopausal women or testosterone in aging men, can lead to accelerated bone loss [57,58]. Former athletes are also at increased risk of RED-S and stress injuries, both of which can contribute to long-term skeletal vulnerability [1,35].

Core treatment includes maintaining adequate calcium (1000–1200 mg/day) and vitamin D (≥30 ng/mL) levels, lifestyle modifications, and pharmacotherapy based on individual fracture risk [1,59]. Pharmacologic treatment in this population is guided by bone mineral density (BMD), fracture history, FRAX scores, and comorbidities.

Bisphosphonates such as alendronate (70 mg weekly), risedronate (35 mg weekly), and zoledronic acid (5 mg IV annually) are first-line therapies due to their ability to significantly reduce vertebral, hip, and non-vertebral fractures [57]. Despite their widespread use, concerns remain regarding the long-term suppression of bone remodeling, especially in aging athletes whose skeletal systems may have experienced repetitive microtrauma or stress from high-impact sports. The reduced flexibility in bone turnover caused by prolonged bisphosphonate use could paradoxically increase the risk of atypical femoral fractures and hinder recovery from skeletal injuries. Although IV zoledronic acid avoids gastrointestinal issues often encountered by athletes with NSAID-induced GI stress, it may still pose challenges regarding adherence and tolerability due to acute phase reactions and concerns about renal safety [1].

Anabolic therapies, including teriparatide (20 mcg SC daily) and abaloparatide (80 mcg SC daily), are indicated in cases of severe osteoporosis or prior vertebral fractures. However, the high cost, limited duration of approved use (typically two years), and potential risks, such as hypercalcemia, raise concerns about their practicality for a physically active, aging population. Additionally, their effectiveness may be restricted for athletes who cannot adhere to strict dosing regimens [1,57]. Romosozumab (210 mg SC monthly), a dual-action monoclonal antibody, offers both anabolic and antiresorptive benefits. While promising in high-risk patients, its use is constrained by emerging data suggesting an elevated risk of cardiovascular events. This could be a particularly relevant concern for aging athletes, many of whom may have underlying or emerging cardiovascular risk factors due to years of intense training or performance-enhancing drug exposure [60].

Denosumab (60 mg SC every 6 months), a RANKL inhibitor, is particularly useful for those with renal insufficiency or bisphosphonate intolerance. It is highly effective in reducing vertebral and hip fractures, but requires strict adherence, as discontinuation can lead to rebound bone loss and multiple vertebral fractures if not followed by another antiresorptive agent [61]. This rebound phenomenon may be especially problematic in aging athletes who may experience interruptions in therapy due to injuries, surgeries, or changing life circumstances.

In male athletes, testosterone deficiency should be evaluated, particularly if clinical signs of hypogonadism are present. Although testosterone replacement therapy (TRT) can enhance bone mineral density (BMD), it is not recommended as a primary treatment for osteoporosis. Additionally, TRT carries risks such as erythrocytosis, increased cardiovascular strain, and the potential suppression of natural testosterone production. In aging athletes, particularly those with a history of anabolic steroid use, TRT should be approached with great caution [58,62]. In postmenopausal women, estrogen replacement may have bone benefits but is reserved for those with additional menopausal symptoms and minimal cardiovascular risk [58].

In patients with very high fracture risk, a sequential approach—starting with anabolic therapy followed by antiresorptives—is often superior to monotherapy [1,57]. Routine monitoring using DEXA every 1–2 years, along with serum calcium, vitamin D, and bone turnover markers, is advised. However, such markers may not fully capture bone quality or resilience in aging athletes, whose unique biomechanical demands and training history warrant critical clinical judgment [59]. A highly individualized approach—encompassing previous fractures, sport-specific stress exposures, and comorbidities—remains essential, especially given the limitations of current pharmacotherapy in preserving the functional skeletal integrity necessary for continued athletic participation in later life.

In athletes, particularly elite competitors (though less so in recreational and older athletes), the use of medications always carries a certain risk of a positive doping test. Doping is a problem even in amateur sports, even though it is less controlled than in professional competitions. Many amateur athletes do use agents to improve their performance, usually not thinking that in amateur sports, they could be tested. As mentioned above, as new antiresorptive drugs are being developed, the regulations also evolve. This is why it is of utmost importance to regularly check the latest WADA Prohibited List. According to the most recent update from the World Anti-Doping Agency (WADA), effective 1 January 2025, bisphosphonates, denosumab, teriparatide, and abaloparatide are not classified as prohibited substances, but some anabolic agents that increase bone growth are. However, athletes should not assume full safety and must consult with their sports governing bodies or anti-doping organizations to ensure compliance with the most current rules. Doping regulations can change over time, and in some cases, specific national or international sports federations may impose additional restrictions beyond WADA’s guidelines. Therefore, staying informed and seeking official clearance before starting treatment is always recommended.

## 6. Study Limitations

This review has several limitations, primarily due to its narrative approach. It lacks predefined search strategies and does not incorporate visual tools to illustrate the selection process of studies and literature. Adopting a systematic review protocol would enhance transparency and improve the reliability of the conclusions drawn by readers.

## 7. Conclusions

Although regular exercise is important for bone and overall health, some types of activities increase the risk of osteoporotic fractures. Preventing injuries while staying active can be a challenge, especially in the population of older competitive recreational athletes performing high volumes of weekly training. Although high levels of physical activity in older adults are usually accompanied by thicker cortical bone and higher physical fitness, the age-related bone loss, hormonal changes, improper nutrition, and previous history of fractures may constitute serious risk factors for osteoporotic fractures. DEXA and regular bone density tests should be used to detect osteoporosis on time so that actions and treatment can start before fractures appear. A DEXA scan should be considered for recreational male athletes starting at age 60 and for females five years after menopause. It is recommended to repeat the scan every 1 to 2 years. Additionally, individuals over 50 years old should receive a DEXA scan if they have a history of fractures, especially low-impact fractures. Together with pharmacological treatment, strength training and stability exercises should be incorporated, especially if the main sport is not high-impact bearing. Strength training exercises that would elicit a positive impact on slowing the unavoidable osteoporosis process are recommended 2 to 3 times a week. If performed only 2 times a week, the exercises should address all major muscle groups, so the training would usually consist of eight to ten exercises for different muscle groups, repeated in 3 to 4 sets of 8 to 10 repetitions. It means that the load is around 75–80% of 1-RM, and the last repetition in the set should be performed with high effort. Good lifestyle choices and calcium and vitamin D complement the approach to the problem, and their use should be based on laboratory findings and diet questionnaires.

## Data Availability

Not applicable.

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
