# Peer review of "Conservative and Pharmacological Strategies for Preventing Osteoporotic Stress Fractures in Older Recreational Competitors"

_healthcare, 2025, doi:10.3390/healthcare13182328_

Round 1
Reviewer 1 Report
Comments and Suggestions for Authors
Overall, the authors address an important topic from both clinical and practical perspectives. While I found the review well-written and informative, there are several points to consider before a final recommendation can be made.
To understand the basis of the review, it is necessary to state the type of review (i.e., narrative) and the selection of information (i.e. selection of studies) considered.
There are several statements that need to be referenced, e.g., “Most people reach their peak bone mass around the age of 30” (ref); or “older athletes often face additional challenges, such as degenerative joint disease, which are alleviated by cycling and swimming” (ref); and several others.
Some parts should be covered in more detail, such as “nutrients”; there you may also consider the intake of protein-rich foods and the restriction of excessive salt, alcohol, and caffeine consumption; etc.
Based on the title, 'Conservative' and 'Pharmacological' strategies should be treated as separate chapters. In particular, conservative strategies are largely lacking.
The conclusion refers to 'strength training and stability exercises'; these topics require further discussion in the text.
Recent studies (reviews and meta-analyses) discussing treatment options for osteoporosis should be included. For example, see the following: PMID: 33183112, PMID: 40469971, etc.
What about considering Chinese Medicine? e.g., doi: 10.3389/fendo.2025.1606753. eCollection 2025.
As with all studies, also this one has limitations. Please include a section on these limitations and provide some remarks on what needs to be done in future.
The text needs proper proofreading and language editing.
Comments on the Quality of English LanguageThe text needs proper proofreading and language editing.
Author Response
Overall, the authors address an important topic from both clinical and practical perspectives. While I found the review well-written and informative, there are several points to consider before a final recommendation can be made.
To understand the basis of the review, it is necessary to state the type of review (i.e., narrative) and the selection of information (i.e. selection of studies) considered.
- Thank you for your comment. We have clarified and emphasized the type of review conducted in the “Method” section, and added a new paragraph specifying the databases that were consulted: “The sources for this review were identified from several databases, including Scopus, Web of Science (WoS), Sport Discus, and PubMed. This article presents a narrative review that synthesizes existing evidence on osteoporotic stress fractures and prevention strategies specifically for older recreational athletes. We prioritized articles published after 2010; however, we also included older studies, particularly those discussing mechanisms of bone remodelling. The goal of this review is to summarize existing knowledge applicable to the older athletic population and to identify their specific challenges and potential therapeutic options. The key terms used in our search were: “osteoporosis,” “stress fracture,” “older athlete,” “recreational athlete”, “bone remodelling,” “antiresorptive agents,” and “fracture prevention.” Most of the selected articles focused on bone health in older athletes or the risk of fractures. This review is descriptive and summative in nature, and no statistical synthesis or original data analysis was performed
In older athletes, the trend of engaging in high-volume and high-intensity recreational activities has only gained momentum in the past two decades. As this population continues to grow, we anticipate that more original research will emerge over time. The current lack of high-quality, original studies specifically focused on older recreational athletes is a major limitation of our review, as it prevented us from conducting a meta-analysis. Future research in this area will be essential to enable more robust quantitative synthesis and evidence-based recommendations". Page 3, line 120-140
There are several statements that need to be referenced, e.g., “Most people reach their peak bone mass around the age of 30” (ref); or “older athletes often face additional challenges, such as degenerative joint disease, which are alleviated by cycling and swimming” (ref); and several others. Some parts should be covered in more detail, such as “nutrients”; there you may also consider the intake of protein-rich foods and the restriction of excessive salt, alcohol, and caffeine consumption; etc.
-Agree, therefore we have added following paragraph: "Although we would expect older athletes to lead and maintain a health-oriented lifestyle, they still underestimate their nutritional needs during periods of high-volume and high-intensity training. Therefore, it is important to regularly evaluate calcium and vitamin D intake, preferably before the conditioning phase of the season and once during the competitive period (typically twice a year for most sports). Based on laboratory findings and nutritional questionnaires, supplementation should be considered as needed."
Page 5, Line 221-227
Based on the title, 'Conservative' and 'Pharmacological' strategies should be treated as separate chapters. In particular, conservative strategies are largely lacking.
- Thank You for pointing this out. We have left the existing text, but it is now separated as a individual chapter. For example, the previous title “Application of Anti-Resorptive/Anabolic Agents in Older Athletes” has become chapter 4. Pages 4-8
The conclusion refers to 'strength training and stability exercises'; these topics require further discussion in the text.
-Thank You for pointing this out, therefore we have added this paragraph accordingly: "Strength training exercises that would elicit a positive impact on slowing the unavoidable osteoporosis process are recommended 2 to 3 times a week. If performed only 2 times a week the exercises should address all major muscle groups, so the training would usually consist of eight to ten exercises for different muscle groups, repeated in 3 to 4 sets of 8 to 10 repetitions. It means that the load is around 75- 80%.” Page 10, paragraph Line 474-476.
What about considering Chinese Medicine? e.g., doi: 10.3389/fendo.2025.1606753. eCollection 2025.
- Thank You for highlighting this paper. While we acknowledge the growing interest in this area, our review focuses on conservative and pharmacological strategies in Western medical practice. Future reviews with Traditional Chinese Medicine could be incorporated.
As with all studies, also this one has limitations. Please include a section on these limitations and provide some remarks on what needs to be done in future.
-Thank You for pointing this out. We have provided text to Your comment accordingly: “In older athletes, the trend of engaging in high-volume and high-intensity recreational activities has only gained momentum in the past two decades. As this population continues to grow, we anticipate that more original research will emerge over time. The current lack of high-quality, original studies specifically focused on older recreational athletes is a major limitation of our review, as it prevented us from conducting a meta-analysis. Future research in this area will be essential to enable more robust quantitative synthesis and evidence-based recommendations". Page 3, line 133-140.
The text needs proper proofreading and language editing.
- Thank You, regarding the language we made changes accordingly.
-----------------------------
Thank you very much for your valuable observations and feedback. We have made several improvements to the manuscript, including changes based on your comments as well as suggestions from another reviewer.
For clarity, we have attached a revised version of the manuscript highlighting all modifications.

Reviewer 2 Report
Comments and Suggestions for Authors
Dear authors,
Thank you for the opportunity to review your manuscript titled Conservative and Pharmacological Strategies for Preventing Osteoporotic Stress Fractures in Older Recreational Competitors. This narrative review addresses a highly relevant and timely clinical and public health topic, given the growing number of physically active older adults and the increasing prevalence of osteoporotic stress fractures. The manuscript is generally well-written and informative, but several issues related to structure, clarity, depth of synthesis, and evidence quality need to be addressed to improve its scientific rigor and suitability for publication.
abstract
The abstract introduces the topic appropriately and highlights the paradoxical relationship between physical activity and stress fracture risk. However, the abstract lacks structure and does not clearly delineate the review type, inclusion criteria, or methodology (e.g., was this a scoping, narrative, or integrative review?). The language should be more concise, and key messages better highlighted. Consider clarifying whether this review includes original analysis or is purely descriptive and summative.
introduction
The introduction effectively outlines the clinical importance of osteoporosis in older recreational athletes and the challenges of balancing exercise benefits with fracture risk. However, the rationale for focusing specifically on recreational competitors (as opposed to all older adults) should be better justified. Additionally, a clearer explanation of the novelty of this review—what gap in the literature it fills—is needed. The term “recreational competitor” requires a more precise operational definition, as it may be interpreted broadly and inconsistently across disciplines.
methods
There is no dedicated methodology section, which is a significant limitation. If the manuscript is intended as a narrative review, the authors should explicitly state this. Even in a narrative review, it is good practice to describe how the literature was selected, searched, and reviewed. Which databases were consulted? What keywords or inclusion/exclusion criteria were used? Without this information, the review risks appearing selective or anecdotal. The lack of methodological transparency undermines the reliability and reproducibility of the review.
main content and thematic organization
The manuscript covers a wide range of relevant topics, including:
-
Age-related bone changes
-
Nutritional influences on bone health
-
The paradox of exercise-induced stress fractures
-
Recovery and low energy availability
-
Pharmacological treatment strategies
While the content is rich, the review would benefit from better organization and integration. Many sections read as isolated informational summaries rather than a cohesive synthesis. For example:
-
The relationship between exercise modality (e.g., running vs swimming) and site-specific BMD changes is described but not critically analyzed.
-
The pharmacological treatment section is overly long and reads more like a drug compendium than a focused analysis of their relevance to older athletes.
-
The discussion of RED-S, hepcidin, and Wnt/β-catenin signaling is interesting but too technically dense and insufficiently contextualized in this specific population.
Consider restructuring the review into clearly defined subsections (e.g., exercise risk-benefit paradox, clinical screening tools, non-pharmacologic strategies, pharmacologic options), each with critical discussion and synthesis. More comparative insight between different studies and evidence levels is needed.
scientific rigor and evidence use
While the manuscript references many appropriate and recent studies, it often lacks critical appraisal of evidence quality. The strength of recommendation or the level of evidence supporting various interventions (e.g., bisphosphonates vs romosozumab) is not discussed. When citing pharmacologic interventions, specify which are supported by randomized trials and which are off-label or based on case reports. Avoid implying clinical consensus where there is still debate.
Some of the physiological mechanisms (e.g., hepcidin’s role in bone health, FOXO3a signaling) are too narrowly presented without clinical correlation, especially considering the target readership may be more clinical than molecular.
discussion and clinical implications
The manuscript identifies an important paradox: high levels of physical activity can both enhance and compromise skeletal integrity in older adults. However, this duality is not adequately explored in terms of clinical decision-making or practical recommendations. How should clinicians screen or monitor older athletes at risk? What training adjustments or thresholds are suggested? Is there a role for wearable monitoring, functional assessments, or targeted imaging?
Furthermore, the implications for clinical guidelines are underdeveloped. For example, the paper notes that osteoporosis screening guidelines exist for postmenopausal women but are lacking for older male athletes. This could be a central argument, but it is not developed or supported with specific recommendations.
conclusion
The conclusion appropriately reiterates the need to balance exercise with bone protection, but it is too general and lacks actionable insights. The recommendation for DEXA screening is appropriate, but the timing, frequency, or thresholds for concern are not specified. Similarly, the mention of strength training is appropriate, but without guidance on dosage, frequency, or modality, the conclusion remains vague. Provide at least a high-level summary of key recommendations based on the review
Author Response
abstract
The abstract introduces the topic appropriately and highlights the paradoxical relationship between physical activity and stress fracture risk. However, the abstract lacks structure and does not clearly delineate the review type, inclusion criteria, or methodology (e.g., was this a scoping, narrative, or integrative review?). The language should be more concise, and key messages better highlighted. Consider clarifying whether this review includes original analysis or is purely descriptive and summative.
- Thank You for your comment. We have clarified and emphasized the type of review conducted in the “Method” section and added a new paragraph specifying the databases that were consulted. Regarding the language we made changes accordingly.
introduction
The introduction effectively outlines the clinical importance of osteoporosis in older recreational athletes and the challenges of balancing exercise benefits with fracture risk. However, the rationale for focusing specifically on recreational competitors (as opposed to all older adults) should be better justified. Additionally, a clearer explanation of the novelty of this review—what gap in the literature it fills—is needed. The term “recreational competitor” requires a more precise operational definition, as it may be interpreted broadly and inconsistently across disciplines.
- Thank You for pointing this out. We agree with this comment. Therefore, we have made changes by adding: “ In this review, the term "recreational competitor" refers to older adults who consistently participate in structured sports training and competitions, even though they are not classified as elite or professional athletes.” Page 2, Paragraph 1, lines 44-46.
- Also added: ” This paper addresses a growing and less-studied population of aging recreational competitors. From both scientific and practical perspectives, this group is often overlooked in discussions surrounding osteoporosis risk management.” Page 2, Paragraph 1, line 52-55.
“The primary goal of this review is to inform and alert all experts and organizers involved in recreation and sporting activities for older participants. It is crucial to recognize that the training practices typically used for younger athletes should not be applied in the same way to this population due to differences in human biology. Additionally, the review emphasizes the importance of effectively managing potential risks.” Page 3, Paragraph 4, line 113-118.
methods
There is no dedicated methodology section, which is a significant limitation. If the manuscript is intended as a narrative review, the authors should explicitly state this. Even in a narrative review, it is good practice to describe how the literature was selected, searched, and reviewed. Which databases were consulted? What keywords or inclusion/exclusion criteria were used? Without this information, the review risks appearing selective or anecdotal. The lack of methodological transparency undermines the reliability and reproducibility of the review.
- Thank You for pointing this comment, therefore we added entire paragraph accordingly: “The sources for this review were identified from several databases, including Scopus, Web of Science (WoS), Sport Discus, and PubMed. This article presents a narrative review that synthesizes existing evidence on osteoporotic stress fractures and prevention strategies specifically for older recreational athletes. We prioritized articles published after 2010; however, we also included older studies, particularly those discussing mechanisms of bone remodeling. The goal of this review is to summarize existing knowledge applicable to the older athletic population and to identify their specific challenges and potential therapeutic options. The key terms used in our search were: “osteoporosis,” “stress fracture,” “older athlete,” “recreational athlete”, “bone remodeling,” “antiresorptive agents,” and “fracture prevention.” Most of the selected articles focused on bone health in older athletes or the risk of fractures. This review is descriptive and summative in nature, and no statistical synthesis or original data analysis was performed. main content and thematic organization.“ Page 3, line 122-132.
While the content is rich, the review would benefit from better organization and integration. Many sections read as isolated informational summaries rather than a cohesive synthesis. For example:
-
The pharmacological treatment section is overly long and reads more like a drug compendium than a focused analysis of their relevance to older athletes.
- We agree with this comment. Therefore, the pharmacological treatment section was modified accordingly. Pages 8-9.
-
The discussion of RED-S, hepcidin, and Wnt/β-catenin signaling is interesting but too technically dense and insufficiently contextualized in this specific population.
- Agree. We have revised and modified the following text. “ Hepcidin is a hormone that regulates iron levels and also contributes to bone metabolism. Typically, hepcidin levels increase after high-intensity exercise, which aids in recovery. Research indicates that in postmenopausal female athletes, hepcidin levels may remain suppressed for extended periods. This delayed recovery of hepcidin can hinder bone formation by disrupting the Wnt/β-catenin pathway, which is crucial for activating osteoblasts and facilitating bone repair. Moreover, the high-impact activities that are intended to enhance bone remodelling during recovery can lead to increased microdamage in the bones. As a result, older athletes may require longer recovery periods to support effective bone remodelling. In our expert opinion, the commonly recommended 48-hour recovery time should be extended to a minimum of 72 hours, avoiding high-impact forces on the bones during this period. Incorporating low magnitude and high frequency (LMHF) mechanical loading through activities such as postural control exercises can be beneficial. These exercises can be safely performed even during recovery from high-impact activities, as they have been shown to have a positive anabolic effect on bones (28). “ Pages 5-6, Line 251-265.
Consider restructuring the review into clearly defined subsections (e.g., exercise risk-benefit paradox, clinical screening tools, non-pharmacologic strategies, pharmacologic options), each with critical discussion and synthesis. More comparative insight between different studies and evidence levels is needed.
- Thank You for pointing this out. We have left the existing text, but it is now separated as a individual chapter. For example, the previous title “Application of Anti-Resorptive/Anabolic Agents in Older Athletes” has become chapter 4. Pages 4-8.
scientific rigor and evidence use
While the manuscript references many appropriate and recent studies, it often lacks critical appraisal of evidence quality. The strength of recommendation or the level of evidence supporting various interventions (e.g., bisphosphonates vs romosozumab) is not discussed. When citing pharmacologic interventions, specify which are supported by randomized trials and which are off-label or based on case reports. Avoid implying clinical consensus where there is still debate.
- We have inserted our own opinion, since there are no works specifically on this topic, they have all been cited.
Some of the physiological mechanisms (e.g., hepcidin’s role in bone health, FOXO3a signaling) are too narrowly presented without clinical correlation, especially considering the target readership may be more clinical than molecular.
- Thank you for pointing this out. We agree with this comment. Therefore, we have made changes in the text together with Your previous comment. Pages 5-6, Line 251-264.
discussion and clinical implications
The manuscript identifies an important paradox: high levels of physical activity can both enhance and compromise skeletal integrity in older adults. However, this duality is not adequately explored in terms of clinical decision-making or practical recommendations. How should clinicians screen or monitor older athletes at risk? What training adjustments or thresholds are suggested?
- According to Your comment we added the entire paragraph supported by additional references: “ The connection between osteoporosis and low energy availability (LEA) and relative energy deficiency in sport (RED-S) is well established. There is a significant variation in the prevalence of LEA/RED-S in different sports (reported prevalence range of 23-79.5% for females and 15-70% for males). However, the highest prevalence is reported for endurance athletes (Gowers 2025 10.1080/15502783.2025.2496448; Mountjoy 2023 10.1136/bjsports-2023-106994). In the studies addressing the energy availability in middle- and long-distance runners, LEA was identified in 43.4% of the athletes (Gallant 2025 10.1007/s40279-024-02130-0). In endurance sports like marathons and long-distance running, the high volume of activity may contribute to low energy availability, leading to RED-S. LEA can thus be unintentional, if the athletes are unaware of the energy demands imposed by their training program, or intentional, if the athletes deliberately restrict their energy intake (Gowers 2025 10.1080/15502783.2025.2496448). Adequate dietary intake for older athletes depends on energy needs from their sport and is influenced by age-related physiological and metabolic changes, along with potential risk factors or chronic illnesses (Guo 2023 10.3390/nu15234973). A recent systematic review indicated that master athletes may not meet their dietary intake requirements necessary for both health and athletic performance (Guo 2023 10.3390/nu15234973). At the 2022 and 2023 World Masters Athletics Championships, it was noted that competitors did not meet the recommended intake of carbohydrates, proteins, vitamins D, E, and K, calcium, potassium, vitamin A, and other nutrients (Leonhardt 2024 10.3390/nu16040564).With regard to bone health, short-term LEA has been linked with increased markers of bone resorption, i.e., decreased markers of bone formation in physically active females, while in men this relationship is less understood (Papageorgiou 2018 10.1007/s00394-017-1498-8). The short-term LEA effect on bone metabolism has, among others, been linked with decreased insulin and leptin levels (Papageorgiou 2018 10.1007/s00394-017-1498-8). As evidenced by cross-sectional studies, long-term LEA in physically active persons was related to lower bone mass, bone resorption, decreased bone strength and increased stress fracture risk (Papageorgiou 2018 10.1007/s00394-017-1498-8). To avoid these negative consequences, the athletes—especially older ones—should have nutritional support to ensure energy availability of at least 30-45 kcal ∙ kg lean body mass-1 ∙ day-1 and the adequate intake of micronutrients essential for bone health, previously identified as deficient in this population, including calcium, vitamin D and K (Papageorgiou 2018 10.1007/s00394-017-1498-8; Leonhardt 2024 10.3390/nu16040564). Page 6, line 260-298.
Is there a role for wearable monitoring, functional assessments, or targeted imaging?
- Agree, we have added: "DEXA scan is recommended for male recreational athletes starting at age 60. For females, it is suggested five years after menopause. Every 1-2 years is a recommended period to repeat the scan to monitor bone density changes. For the athletes over the age of 50, a DEXA scan should be considered if there is a history of fractures, particularly low-impact ones". Page 8, paragraph 1, line 349-354.
Furthermore, the implications for clinical guidelines are underdeveloped. For example, the paper notes that osteoporosis screening guidelines exist for postmenopausal women but are lacking for older male athletes. This could be a central argument, but it is not developed or supported with specific recommendations.
- Thank You for pointing this out. We have added the following text concerning older male athletes: “Although testosterone replacement therapy (TRT) can enhance bone mineral density (BMD), it is not recommended as a primary treatment for osteoporosis. Additionally, TRT carries risks such as erythrocytosis, increased cardiovascular strain, and the potential suppression of natural testosterone production. In aging athletes, particularly those with a history of anabolic steroid use, TRT should be approached with great caution. (56, 50)". Page 9, paragraph 3, line 419-424.
conclusion
The conclusion appropriately reiterates the need to balance exercise with bone protection, but it is too general and lacks actionable insights. The recommendation for DEXA screening is appropriate, but the timing, frequency, or thresholds for concern are not specified.
- We agree, therefore changes have been made: “A DEXA scan should be considered for recreational male athletes starting at age 60 and for females five years after menopause. It is recommended to repeat the scan every 1 to 2 years. Additionally, individuals over 50 years old should receive a DEXA scan if they have a history of fractures, especially low-impact fractures.” Page 9, paragraph, line 428-436.
Similarly, the mention of strength training is appropriate, but without guidance on dosage, frequency, or modality, the conclusion remains vague. Provide at least a high-level summary of key recommendations based on the review.
- Thank You for posting this out, therefore we have added this point accordingly: "Strength training exercises that would elicit a positive impact on slowing the unavoidable osteoporosis process are recommended 2 to 3 times a week. If performed only 2 times a week the exercises should address all major muscle groups, so the training would usually consist of eight to ten exercises for different muscle groups, repeated in 3 to 4 sets of 8 to 10 repetitions. It means that the load is around 75- 80%. Also, We modified the following sentence ”. Page 10, paragraph, line 474-476.
-----------------------------------
Thank you very much for your valuable observations and feedback. In response, we have made several improvements to the manuscript, including changes based on Your comments as well as suggestions from another reviewer.
For clarity, we have attached a revised version of the manuscript highlighting all modifications.
